# Protein Targeting to Glycogen (PTG): A Promising Player in Glucose and Lipid Metabolism

**DOI:** 10.3390/biom12121755

**Published:** 2022-11-26

**Authors:** Xia Deng, Chenxi Wang, Yue Xia, Guoyue Yuan

**Affiliations:** Department of Endocrinology, Affiliated Hospital of Jiangsu University, Zhenjiang 212001, China

**Keywords:** protein targeting to glycogen (PTG), protein phosphatase 1α (PP1α), glucose metabolism, lipid metabolism, glycogen synthesis

## Abstract

Protein phosphorylation and dephosphorylation are widely considered to be the key regulatory factors of cell function, and are often referred to as “molecular switches” in the regulation of cell metabolic processes. A large number of studies have shown that the phosphorylation/dephosphorylation of related signal molecules plays a key role in the regulation of liver glucose and lipid metabolism. As a new therapeutic strategy for metabolic diseases, the potential of using inhibitor-based therapies to fight diabetes has gained scientific momentum. PTG, a protein phosphatase, also known as glycogen targeting protein, is a member of the protein phosphatase 1 (PP1) family. It can play a role by catalyzing the dephosphorylation of phosphorylated protein molecules, especially regulating many aspects of glucose and lipid metabolism. In this review, we briefly summarize the role of PTG in glucose and lipid metabolism, and update its role in metabolic regulation, with special attention to glucose homeostasis and lipid metabolism.

## 1. Introduction

Carbohydrates and lipids are the main sources of energy required by organisms to maintain life activities. The homeostasis imbalance of the regulatory network is also an important inducing factor for the occurrence and development of a variety of metabolic diseases [1,2,3]. Glucose and lipid metabolism disorder is the main clinical phenotype of obesity, type 2 diabetes and nonalcoholic fatty liver disease, and can also induce cardiovascular and cerebrovascular diseases [4]. Under the influence of genetic and/or environmental factors, the decrease of insulin secretion or the disturbance of signal transduction (insulin resistance) will lead to the decrease of energy supply from glucose sources and the enhancement of lipid decomposition and release, and can stimulate liver gluconeogenesis, promote liver glucose release, and participate in the occurrence and development of diabetes and its complications [5,6]. In addition, hormones such as estrogen, cortisol, and insulin, and excess nutrition, can promote the liver to synthesize more triglycerides, and excess glucose can also be transformed into triglycerides through a lipogenesis reaction [7]. These excessively synthesized triglycerides accumulate in the liver, which may lead to the formation of nonalcoholic fatty liver disease [3].The disorder of liver lipid metabolism can cause liver steatosis, obesity, insulin resistance, and can aggravate the process of diabetes [8,9].With the rapid increase of the prevalence of these metabolic diseases in the world, it is more and more important to explore the regulation of glucose and lipid metabolism and its function in the homeostasis.

The liver plays an important role in various organs involved in glucose and lipid metabolism [10]. The homeostasis of intracellular glucose and lipid mainly depends on the liver [11,12]. Glycogen and lipid are the main forms of energy storage, which are strictly regulated by hormones and metabolic signals and provide energy at different stages [13]. When the postprandial blood glucose concentration increases, glucose is firstly phosphorylated to glucose 6-phosphate after it is transported to hepatocytes with blood [14]. Glucose 6-phosphate can no longer penetrate the cell membrane and return to the blood. Instead, it acts as the source of carbohydrate metabolism in the liver, opening up metabolic pathways such as glycogen synthesis, glycolysis, pentose phosphorylation and lipogenesis. Under starvation, hepatocytes can also convert non-glucose substances such as lactic acid and glycogenic amino acids into glucose-6-phosphate through the process of gluconeogenesis [15,16]. The liver is one of the two major sites for the conversion of carbohydrates into fats in the human body, and is also the regulatory center for the digestion, absorption, synthesis and transportation of lipids [10]. It promotes the digestion and absorption of lipids by secreting bile, and is the most important place for the synthesis of lipids such as triglycerides, fatty acids, cholesterol, and ketones (Figure 1). Under normal circumstances, these lipid metabolites are transported by the blood to extrahepatic tissues for storage or utilization after being synthesized in the liver [7,17].

Phosphorylation and dephosphorylation of proteins are widely recognized as key regulators of cellular function [18,19] and are often referred to as “molecular switches” in the regulation of cellular physiological processes [20,21]. Protein phosphorylation and dephosphorylation can regulate signal transduction in a variety of ways, including by activating or inhibiting conformational changes to create binding sites for proteins containing specific domains, thereby affecting the conformation of proteins or protein–protein interactions, and controlling their cellular localization [22,23]. Numerous studies have shown that the phosphorylation/dephosphorylation of related signaling molecules plays a key role in the regulation of hepatic glucose and lipid metabolism [24,25,26,27]. Glycogen targeting protein (PTG) is a member of the protein phosphatase 1 (PP1) family and can play a role by catalyzing the dephosphorylation of phosphorylated protein molecules. Previous studies have shown that it can affect glycogen metabolism in liver and skeletal muscles by regulating the dephosphorylation of ribosomal S6 protein kinase, glycogen synthase, and glycogen phosphorylase [28,29]. This article reviews the research progress of protein targeting to glycogen (PTG) in glucose and lipid metabolism in recent years, and provides new insights for the prevention and treatment of metabolism-related diseases and clinical transformation research.

## 2. Structure and Characteristics of PP1

Protein phosphorylation is an important cellular regulatory mechanism, and many enzymes and receptors are activated/deactivated by phosphorylation and dephosphorylation events through kinases and phosphatases [22,23]. It is generally believed that phosphorylation or dephosphorylation of proteins is one of the initial steps in the coordination of cell and organ functions, such as regulation of glycolipid metabolism, cell proliferation, apoptosis, inflammation, and other important physiological processes. Protein kinases and protein phosphatases catalyze this process [24,25,26,27]. Therefore, kinases and phosphatases are promising targets for regulating this reaction, and many inhibitors against these proteins have been developed [30]. Krzyzosiak et al. [31] discovered Raphin1, a serine/threonine phosphatase inhibitor that mainly inhibits PPP1R15B, which is reported to be a potential drug candidate for the treatment of Huntington’s disease. Furthermore, some studies have pointed out that PTP1B inhibitors are considered to have obvious antidiabetic potential [32]. Therefore, the study of protein phosphatase is of great significance.

The PP1 regulatory subunit, also known as glycogen targeting regulatory subunit, coordinates glycogen synthesis by targeting the catalytic subunit of protein phosphatase 1 to glycogen particles [33,34,35]. They together constitute the PP1 holoenzyme, which regulates the activity of glycogen metabolizing enzymes, and plays a role through PP1 catalyzing the dephosphorylation of phosphorylated protein molecules, mainly dephosphorylating glycogen synthase phosphatase (GSP) and activating glycogen synthase (GS), which in turn stimulates glycogen production [33,36]. According to the GenBank database, there are seven genes encoding G subunit (PPP1R3C-PPP1R3G), all of which have PP1 binding domain and glycogen binding domain. Previous studies have found that many subtypes of ppp1r (PPP1R3A, B, G) can promote glycogen storage when overexpressed in hepatocytes [37,38]. PTG is a subtype of the protein phosphatase 1 family, encoding protein phosphatase regulatory subunit 3C, which can play a role by catalyzing dephosphorylation of phosphorylated protein molecules [39]. The PTG gene is located on human chromosome 10 and mouse chromosome 19. The full length of its cDNA is 1499 BP, encoding 285 amino acid residues [28]. It was found that PTG was expressed in muscle, liver, adipose tissue, heart, brain and other tissues of mice, especially highly expressed in liver and muscle, and has a variety of important biological roles [28,29,40,41]. Previous studies have found that PTG is also involved in the occurrence and development of colorectal cancer [42], and renal cell carcinoma [43].

At present, the prevalence of related diseases caused by disorders of glucose and lipid metabolism is increasing substantially, and a better understanding of the regulation of glucose and lipid metabolism is essential for the prevention and treatment of related diseases and the improvement of people’s quality of life. In recent years, many researchers have conducted in-depth research on the role of PTG in glucose and lipid metabolism [29,40,44].

## 3. Overview of PTG and Its Regulatory Factors

Studies have shown that PTG was mainly expressed in insulin-sensitive tissues such as liver and muscle [39], and the expression of PTG in the liver of mice fed with a high-fat diet was significantly increased [45]. More interestingly, Munro et al. [46] found that the phosphatase activities related to PP1 and PTG were down-regulated by 60–70% in streptozotocin-induced diabetes and recovered through insulin treatment, indicating that the phosphatase activities such as PTG were regulated by insulin. Allaman et al. [47] found that both norepinephrine and adenosine can up-regulate PTG mRNA levels in astrocytes and hepatocytes, while increasing glycogen synthesis. Mapping the major phosphorylation sites of PTG by mass spectrometry analysis, Vernia et al. [48] found that phosphorylation activation of AMPK Ser-8 and Ser-268 increases the formation of a complex of dual-specificity phosphatase and E3 ubiquitin ligase and down-regulates PTG activity, resulting in a decrease in its glycogen activity. This study further confirmed that AMPK can physically interact with PTG and change its basal phosphorylation state. Cheng et al. [41] found multiple forkhead box protein A2 (FoxA2) binding sites in the promoter region of PTG, and further constructed a luciferase reporter gene to confirm that FoxA2 transactivates the PTG promoter in H4IIE liver cancer cells. Likewise, nuclear extracts from mouse liver and H4IIE cells were able to bind FoxA2-specific probes derived from the PTG promoter region. Chromatin immunoprecipitation experiments further demonstrated that FoxA2 could bind to the PTG promoter in vivo. Furthermore, adenosine-3’,5’-cyclic monophosphate (cAMP) treatment also activated the PTG promoter and significantly increased the level of PTG in H4IIE cells. In general, a variety of glucose metabolism-related signaling molecules are involved in the regulation of PTG.

## 4. PTG and Glucose Metabolism

The liver is essential for maintaining normal glucose homeostasis—it produces glucose when fasting and stores it after meals. The maintenance of this homeostasis is multifaceted, mainly through the regulation of gluconeogenesis, glycogen synthesis, glycogenolysis, and glycolysis [13]. The liver has a remarkable ability to produce glucose, which is released into the systemic circulation and utilized by other tissues. Glucose production in the liver results from glycogenolysis and de novo synthesis of glucose (gluconeogenesis). Normally, the production and uptake of glucose are balanced to maintain glucose homeostasis in human body [10]. After a meal, the pancreas β cells secrete postprandial insulin, and the increased insulin signal activates AKT in the cells. Subsequently, glycogen synthase kinase-3β (GSK-3β) is phosphorylated and inactivated, while glycogen synthase is activated, inhibiting the decomposition of glycogen, promoting the synthesis and storage of glycogen, and finally reducing blood glucose levels [49,50]. Additionally, glucagon and adipose free fatty acid release were also indirectly inhibited, resulting in the inhibition of the gluconeogenesis process [51]. Conversely, during fasting, pancreatic α cells can secrete glucagon to act on liver tissue, inactivating glycogen synthase, thereby inhibiting glycogen synthesis. During short-term fasting, glycogenolysis is the main source of glucose released into the blood, and stored glycogen is further hydrolyzed into glucose to increase glucose production in the liver. However, during prolonged periods of fasting, glycogen stores are gradually depleted and glycogenolysis decreases [13]. Under the action of phosphoenolpyruvate carboxy kinase (PEPCK) and glucose-6-phosphatase (G6Pase), non-carbohydrate precursors can resynthesize glucose in the liver, known as gluconeogenesis [52]. In addition, insulin levels also decreased, and the proportion of peripheral tissues using glucose as fuel decreased to maintain blood glucose levels [53]. With the prolongation of fasting time, the contribution of gluconeogenesis to hepatic glucose production gradually increased to increase blood glucose [11].

### 4.1. PTG and Glycogen Synthesis

Glycogen synthesis is dynamically regulated by insulin through synergistic dephosphorylation of the serine/threonine phosphatase PP1 and the glycogenolytic enzyme glycogen phosphorylase (GP) on glycogen synthase (GS), leading to the activation of GS and the deactivation of GP [54]. In the liver, PTG and PPP1R3B(GL)are expressed at roughly equivalent levels [55], and they jointly promote hepatic glycogen mobilization and storage. PTG overexpression significantly increased glycogen content, mainly due to its ability to promote the redistribution of PP1 and glycogen synthase to glycogen granules, significantly increasing GS activity and glycogen synthesis (Figure 2) [28,29]. Studies also noted that PTG recruited phosphatases to where glycogen synthesis occurs, allowing the activation of glycogen synthase and deactivation of glycogen phosphorylase, thereby increasing glycogen synthesis and reducing its degradation [55]. Further study also found that overexpression of the PP1 subunit protein targeting PTG could significantly increase the level of cell glycogen. At the same time, on the fourth day after PTG knockdown, the glycogen level decreased by more than 85%, which reduced the glycogen targeted PP1 protein level, and correspondingly reduced the activity of glycogen synthase and phosphorylase targeted PP1 in cells, and the glycogen decomposition rate increased significantly under PTG knockdown, indicating that PTG mainly played a role in inhibiting glycogen decomposition [56]. This notion is supported by the results of Crosson et al. [44], in which mice with heterozygous deletion of the PTG gene had reduced glycogen storage in adipose tissue, liver, heart and skeletal muscle. Accordingly, the activity of glycogen synthase and the rate of glycogen synthesis decreased. Furthermore, Greenberg et al. [57] found that overexpression of PTG in 3T3-L1 adipocytes discretely stimulated the activity of PP1 on glycogen synthase and phosphorylase, resulting in increased uptake and storage of glucose as glycogen. Similarly, Printen et al. [29] found that PTG also forms complexes with phosphorylase kinase, phosphorylase a and glycogen synthase in 3T3-L1 adipocytes. The latter are the main enzymes involved in the hormone regulation of glycogen metabolism. Further studies by Lu et al. [45] found that feeding mice with a high-fat diet could increase hepatic glycogen, which was due to the increased expression of the glycogen scaffolding protein PTG. Activation of mTORC1 and its downstream target SREBP1 resulted in increased PTG promoter activity and increased glycogen levels in mice and cells. In contrast, deletion of the PTG gene in mice prevented HFD-induced hepatic glycogen accumulation. López-Soldado et al. [14] further bred liver PTG-overexpressing mice, which could maintain relatively high hepatic glycogen levels even when fasting, thereby maintaining hepatic energy status. Some researchers hybridized Akita mice with PTG overexpressing mice in the liver to construct Akita PTG ^OE^ mice with increased liver glycogen content. The blood glucose level of Akita PTG ^OE^ mice decreased gradually with the progress of the disease. Akita PTG^OE^ mice have stronger glycogen production ability, which shifts blood glucose to glycogen synthesis, so the above results suggest that long-term enhancement of hepatic glycogen may have the effect of reducing hyperglycemia [58]. It can be seen that PTG, as a protein phosphatase with dephosphorylation, plays a very important role in regulating energy metabolism in the liver.

PTG also plays an important role as a regulator in the regulation of glycogen metabolism in cells other than the liver. Researchers found that the gene encoding PTG is abundantly expressed in astrocytes of the central nervous system, and some evidence suggests that PTG plays an important role in the regulation of glycogen metabolism in this cell line [59]. Allaman et al. [47] found that in astrocytes, overexpression of PTG resulted in a 100-fold increase in glycogen, a downregulation of endogenous PTG expression by siRNA resulted in a two-fold decrease in glycogen, and knockdown of PTG expression (PTG-KO astrocytes) resulted in an 80 % decrease in glycogen. Interestingly, the reduction in glycogen content measured in PTG-KO astrocytes (80%) was similar to that observed in the brains of PTG-KO mice (70%), further indicating the close parallel relationship between glycogen metabolism in cultured astrocytes and brain glycogen metabolism. These observations suggest that PTG is also a master regulator of astrocyte glycogen, a central element of glycogen regulation in vivo. Furthermore, changes in PTG expression in the brain have been shown to be associated with different behavioral conditions related to glycogen mobilization, such as sleep deprivation and learning and memory [60,61,62]. Astrocytes are the most widely distributed type of cells in the mammalian brain and the largest type of glial cells. It is also the main storage site of glycogen in the brain. When neurons are highly active and the blood glucose provided through the blood brain barrier cannot meet the needs, glycogen in glial cells can be decomposed into glucose to provide energy for neurons under the effect of neurotransmitters [63]. In cultured astrocytes isolated from a mouse model of schizophrenia, a strong reduction in PTG expression was also observed, which is associated with altered glycogen metabolism [59]. The above results suggest that PTG can control brain glycogen metabolism under physiological and pathological conditions (such as psychiatric disorders), thereby controlling the key role of astrocyte function.

A study of skeletal muscle glycogen levels showed that an overexpression of interferon regulatory factor 4 (IRF4) in mice exhibited decreased exercise capacity and decreased glycogen content. Furthermore, PTG expression increased in the absence of IRF4, and decreased when IRF4 was overexpressed. This indicates that IRF4 regulates the glucose metabolism of skeletal muscle by regulating PTG, and the overexpression of IRF4 leads to the low expression of PTG, thereby reducing the content of glycogen in skeletal muscle. However, opinions on the regulation of muscle glycogen levels by PTG are divided. Studies have shown that heterozygous PTG deletion does not result in impaired glycogen synthesis and reduced glycogen levels in the gastrocnemius muscle. This study also found that PTG in C2C12 myoblasts did not significantly change the content of glycogen. These results indicate that the effect of PTG on skeletal muscle glycogen depends on IRF4 [64].

### 4.2. PTG and Gluconeogenesis

The liver plays a major role in maintaining normal glucose homeostasis by controlling the balance between hepatic glucose production and storage [15]. During fasting, an increase in circulating glucagon triggers the gluconeogenesis by activating the cAMP pathway [65]. It has been reported that norepinephrine could stimulate PTG expression in mouse cortical astrocytes via the cAMP pathway [66]. PTG was significantly up-regulated during 3T3-L1 adipocyte differentiation, which involved the cAMP signaling pathway [67]. FoxA2, a transcriptional regulator of hepatic gluconeogenesis genes, mediates cAMP-stimulated PTG transcription by binding to promoters in hepatocytes [41]. Recent studies have found that the phosphorylation/dephosphorylation of related signaling molecules plays a key role in the regulation of hepatic gluconeogenesis, which is mainly regulated by phosphorylase and dephosphorylase. Various hormones can regulate glucose metabolism by affecting the phosphorylation and dephosphorylation status of liver enzymes [68]; glucagon activates adenylate cyclase to produce cAMP, which, on the one hand, activates the cAMP-dependent protein kinase, which causes the phosphorylation of pyruvate kinase to inhibit the effect of pyruvate kinase, and on the other hand, it can also promote the phosphorylation of fructose 2,6-bisphosphatase, which in turn inhibits the glycolytic pathway, stimulates gluconeogenesis, and promotes glucose production, while insulin exhibits the opposite effect [6,69]. In addition, glucagon can also activate peroxisome proliferator-activated receptor gamma coactivator 1α (PGC-1α) to dephosphorylate. Dephosphorylated PGC-1α will be transferred to the nucleus and combined with hepatocyte nuclear factor 4α (HNF4α) to form a complex to activate it. In this way, the transcription of G6Pase and PEPCK-encoding genes is initiated, which affects gluconeogenesis [65,69]. Previous studies also found that glucagon can promote the nuclear translocation and stability of FOXO1 through cAMP-dependent and protein kinase α-dependent phosphorylation of FOXO1, thereby affecting gluconeogenesis [70]. Uebi et al. [71] found that PP1 can dephosphorylate and activate the Ser171 site of CREB-regulated transcriptional coactivator 2 (TORC2). Phosphorylated TORC2 remains inactive in the cytosol, while dephosphorylated TORC2 is translocated to the nucleus. It binds with phosphorylated cAMP response element binding protein (CREB) to form a CREB-TORC2 complex, which increases the expression of key gluconeogenesis enzymes such as G6Pase and PEPCK. Qi et al. [72] found that follicle-stimulating hormone can promote the membrane translocation of G protein-coupled receptor kinase 2 (GRK2), resulting in the phosphorylation of serine 485 of AMPK α and the dephosphorylation of threonine 172. In the nucleus, it promotes gluconeogenesis, while TORC2 translocates into the nucleus and promotes gluconeogenesis. As a member of the PP1 family, PTG has a dephosphorylation effect, suggesting that PTG may be closely related to hepatic gluconeogenesis. Ji et al. [73] found that overexpression of PTG in primary mouse hepatocytes or wild-type mouse liver promoted hepatic glucose production and expression of gluconeogenesis genes. Conversely, PTG knockout reduced hepatic gluconeogenesis and suppressed cAMP-stimulated gluconeogenic gene expression and TORC2 dephosphorylation. Animal studies showed that PTG knockdown in the liver of db/db mice significantly improved blood glucose levels and reduced the expression of key genes of gluconeogenesis (Figure 2). However, other researchers constructed PTG^OE^ hybridized mice with PTG overexpression. The results showed that, compared with normal mice, PTG^OE^ mice showed decreased liver gluconeogenesis and increased glycolysis, and PGC1 α and PEPCK expression levels decreased, further reducing the blood glucose level of the mice [58]. The possible reason for the difference is that, first of all, the models of the two research teams are from different strains of mice, and the differences in genetic background may lead to abnormal results. Secondly, the methods of overexpression and knockdown of PTG adopted by the two groups were inconsistent, which may also be the reason for the difference in the results. In addition, the influence of living environment and intestinal flora on the results cannot be ruled out. Previous studies have shown that hepatic glycogen synthesis is closely related to the process of gluconeogenesis, and that, in addition to glucose uptake, hepatic gluconeogenesis flux also determines the amount of glycogen formed, especially in the fasted state [1,74,75]. Overexpression of PTG in the liver stimulates glycogen synthesis mainly from gluconeogenic precursors through an indirect pathway. The roles and mechanisms of PTG in gluconeogenesis have not been fully elucidated, and more in-depth studies are still needed.

## 5. PTG and Lipid Metabolism

Liver is the central organ that controls lipid homeostasis by precisely regulating various biochemical, signaling and cellular pathways. Hepatic de novo lipogenesis is a fundamental biosynthetic pathway in the liver that helps hepatocytes store synthetic lipids [76]. This process is an extension of the complex metabolic network within the liver, providing substrates primarily through glycolysis and carbohydrate metabolism. Long-term unhealthy lifestyle and overnutrition can lead to disorders of hepatic lipid metabolism, leading to serious lipid-related diseases, including obesity, NAFLD, and T2DM [77,78].

### 5.1. PTG and Fatty Acid Metabolism

The liver is the main site of fatty acid metabolism and the main source of ketogenesis. Fasting will increase the synthesis of TG in the liver, and the promotion of liver steatosis is due to excessive intake of circulating fatty acids produced by fat decomposition of adipose tissue [79]. It was found that prolonged fasting led to liver TG level, plasma β- hydroxybutyric acid, FFA concentration and plasma FGF21 level being significantly increased in the normal feeding of mice, but the increase was significantly reduced in PTG overexpression mice. In the further investigation, prolonged fasting also promoted the expression of the fatty acid metabolism related genes HMGCS2, CD36, PPAR α, CPT1 α and FGF21, while PTG overexpression significantly weakened this effect [14]. This suggests that PTG may be involved in fatty acid uptake or oxidation in the liver. Further research needs to be carried out to clarify this metabolic effect.

### 5.2. PTG and Fat Synthesis

The main feature of liver fat metabolism disorder is excessive accumulation of TG in the liver [80]. Under normal circumstances, the liver only stores a small amount of fatty acids as triglycerides. However, in addition to increasing the intake of free fatty acids, increased fat synthesis can also lead to liver steatosis [81]. In the case of over nutrition and obesity, the metabolism of fatty acids in the liver changes, which usually leads to the accumulation of triglycerides in the liver cells, resulting in liver steatosis and various metabolic related diseases, such as nonalcoholic fatty liver disease, insulin resistance, and T2DM [3,77]. Studies have shown that circulating hyperglycemia enters the liver and is converted into glycogen for storage. When there is too much glycogen, glucose enters the glycolytic pathway, and the pyruvate and acetyl coenzyme A produced will generate malonyl coenzyme A under the action of ACC, and then FAS catalyzes the production of long-chain saturated fatty acids to promote fatty acid synthesis [12]. Moreover, glucose can also promote the expression of SREBP-1 [82], which can up-regulate the expression of ACC and FAS genes and increase the synthesis of fatty acids [83]. Previous studies have found that PTG mainly plays an important regulatory role in glycogen synthesis (Figure 2) [28,29,40]. The latest evidence has shown that PTG-silencing mice can prevent the accumulation of liver glycogen induced by high fat, reduce fasting blood glucose and insulin levels, and improve insulin sensitivity; mTORC1/SREBP1 can act on PTG promoter, regulate PTG transcription and affect glycogen metabolism, and glycogen accumulation can regulate SREBP1 expression through feedback [45]. More interestingly, the study by Wang et al. [84] found that AMPK inhibits mTORC1 activity by phosphorylating mTORC1 pathway members TSC2 and Raptor, thereby preventing protein and lipid synthesis. In the adipose tissue of mice PTG was overexpressed, using the fatty acid binding protein 4 (FABP4) promoter, and the results showed that glycogen levels in the adipose tissue were 200 to 400 times higher than in wild-type mice, which shows that adipocytes have the steric capacity to accommodate high levels of glycogen (Figure 2), and the importance of glycogen in fat deposition is demonstrated. The study found that transgenic overexpression of PTG in adipose tissue increased glucose flux into the glycogen synthesis pathway, indicating that adipocytes are capable of storing high levels of glycogen [54]. Cui et al. [85] found that PTG can promote the adipocyte differentiation of 3T3-L1 preadipocytes, and found that knockdown of SREBP1 gene expression led to down-regulation of PTG gene expression, speculating that the effect of PTG on adipogenesis may be regulated by SREBP1 in 3T3-L1 to be realized. Lu et al. [45] found that PTG silencing in mice could prevent high-fat-induced accumulation of hepatic glycogen, reduce hepatic lipid deposition, reduce fasting glucose and insulin levels, and improve insulin sensitivity, while Rapamycin complex 1 (mTORC1) and cholesterol regulatory element binding protein 1 (SREBP1) can act on the PTG promoter to regulate the transcription of PTG and affect the metabolism of glycogen. Moreover, the accumulation of glycogen can regulate SREBP1 through feedback, thereby affecting fat metabolism, establishing a dialogue between hepatic glucose and lipid metabolism, and regulating energy balance, indicating that PTG plays an important role as a bridge in glycogen and lipid energy balance. It was found that the reduced liver adipogenesis in normal Akita mice was restored in Akita PTG^OE^ mice, and the expression of FAS and SCD1, the key regulatory factors of adipogenesis, was up-regulated. Increased lipogenesis may be due to increased hepatic glucokinase, as glucokinase overexpression has been reported to lead to increased hepatic glycolytic flux, resulting in increased concentrations of glycerol-3-phosphate and malonyl-CoA, the latter of which are a nascent fat generated substrate. More importantly, Akita PTG^OE^ mice had reduced hepatic lipolytic capacity. Thus, the increase in hepatic TG observed in Akita PTG^OE^ mice was associated with higher lipogenesis and lower lipolysis in the liver [58].

## 6. Conclusions

In conclusion, PTG is a pleiotropic and promising protein phosphatase which seems to play an important role in every link of glucose and lipid metabolism, and plays different key roles in different physiological conditions and pathological abnormalities. It may be an important protein that leads to the interaction of abnormal liver glucose and lipid metabolism. PTG may become a new therapeutic target for metabolic diseases in the near future, with great clinical value. Of course, it is worth further exploring gluconeogenesis and lipid metabolism and its clinical application in transformation.

## Figures and Tables

**Figure 1 biomolecules-12-01755-f001:**
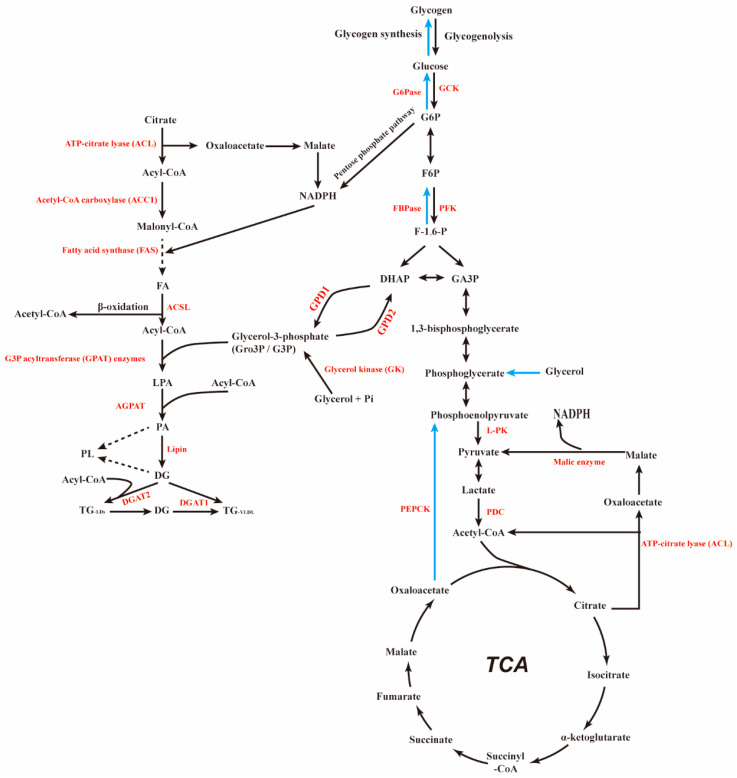
The enrichment of glucose-induced glucokinase (GCK), which phosphorylated glucose to glucose-6-phosphate (G-6-P); it is used as a substrate for glycolysis or glycogen synthesis, depending on the nutritional status. The product pyruvate is further decarboxylated to acetyl coenzyme A and enters the tricarboxylic acid cycle (TCA) in mitochondria. Acetyl coenzyme A or malonyl coenzyme A synthesize fatty acids from scratch and further process them into TAG. In mammals, FA synthesis is catalyzed by acetyl CoA carboxylase (ACC) and fatty acid synthase (FAS).

**Figure 2 biomolecules-12-01755-f002:**
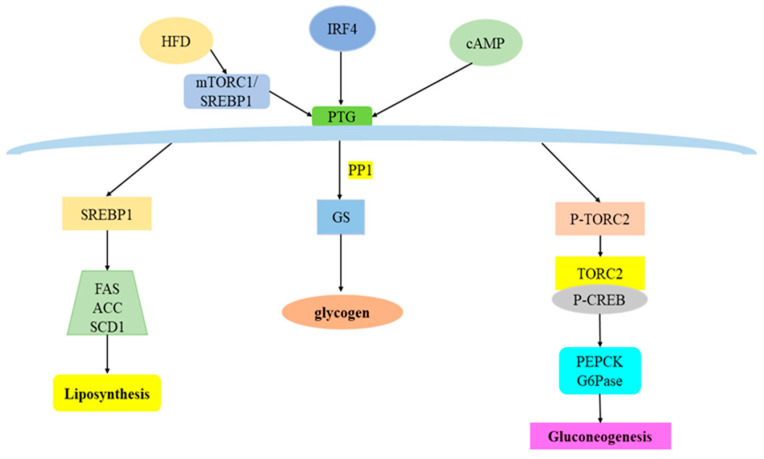
Possible PTG mediated crosstalk mechanism of glucose and lipid metabolism. The effect of PTG on fat metabolism is still controversial, but it can promote fat synthesis and promote the expression of related genes, such as SREBP1, FAS, SCD1, ACC. In addition to influencing glycogen synthesis, recent studies also found that PTG can promote the expression of PEPCK and G6Pase, key enzymes of gluconeogenesis, and then participate in gluconeogenesis.

## Data Availability

Not applicable.

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
