# Peer review of "Protein Targeting to Glycogen (PTG): A Promising Player in Glucose and Lipid Metabolism"

_biomolecules, 2022, doi:10.3390/biom12121755_

Round 1

Reviewer 1 Report

This is a rather interesting review of the role of PTG in glucose and lipid metabolism.

I have comments and suggestions:

1. The authors abuse with the connector "in addition". The authors may reformulate your phrases to avoid it or use "moreover" or "furthermore" instead.

2. Line 86 and 91. It is not clear what the authors mean by the abbreviation GS.

3.  It would be better to use the text "Line 110-112@ in the introduction section.

4. There is no definition for "GL" line 171.

5. In the chapter "4. PTG and glucose metabolism". There is no information about PTG or it is not clear.

6. There is no semantic connection between sentences in the lines 222-223.

The authors may use a new paragraph for it.

Author Response

1.The authors abuse with the connector "in addition". The authors may reformulate your phrases to avoid it or use "moreover" or "furthermore" instead.

Response: Thank you for your advice. Relevant modifications have been made to the conjunctions in the manuscript (line 83,87,126,142,158,186,276,294,334,349,367).

  1. Line 86 and 91. It is not clear what the authors mean by the abbreviation GS.

Response: I'm sorry for this inappropriate statement. "(G subunit)" does not refer to the abbreviation GS, but another expression of glycogen targeting regulatory subunit (line 91).

3.It would be better to use the text "Line 110-112@ in the introduction section.

Response: Thank you very much for your suggestion and we have moved it to the introduction section (Lines 68 to 71).

  1. There is no definition for "GL" line 171.

Response: We are very sorry for the misunderstanding caused by the format error. PP1 has four glycogen targeting subunits: GM, GL, PPP1R6 and glycogen target protein (PTG). GL is also known as PPP1R3 (line 177).

  1. In the chapter "4. PTG and glucose metabolism". There is no information about PTG or it is not clear.

Response: In the chapter "4. PTG and glucose metabolism", the first paragraph introduces the background knowledge related to glucose metabolism, so as to describe the relationship between PTG and different glucose metabolism pathways later.

  1. There is no semantic connection between sentences in the lines 222-223.

Response: Indeed, these sentences lack cohesion in reading, so the content of this part will be segmented after consideration (line 240-241).

Reviewer 2 Report

The review entitled “PTG: the molecular link between glycolipid metabolism” by Deng et al clearly summarises the function and role of PTG and lipid metabolism regulation. I think this is a well-written review and will be of interest to the field. As such I have minor comments listed below:

Comments:

·         In the introduction, you give a detailed overview of glycogen metabolism, however, I feel this information would be better supported with a figure describing the pathways.

·         Line 76-78, this statement is a little confusing, and I feel following on from the previous sentence it would be better to state “As such the kinases and phosphatases are promising targets to modulate this response and such many inhibitors targeting these proteins have been developed”. I also think there is a lot of excellent points made about these inhibitors, and this information should also be summarised in a table to make it clear to the reader what they are targeting. This type of table would be very useful for the field.

·         Line 85 Protein phosphatase 1 should have the abbreviation after it (this happens in the second sentence.

·         Line 208- You mention that PTG plays a role in glycogen synthesis in astrocyte cells. Is this specific to astrocyte cells in the CNS? Or are their other CNS cell types that PTG plays a role? If astrocytes are special in this sense, I think an explanation of what is currently know about astrocyte metabolism would be helpful- as they have alternate metabolism to neurons for example.

·         Line 221- there is a reference missing

·         Line 223- I would make this a separate paragraph

·         I think it’s interesting that there is a role for IRF4 and cGAMP in PTG expression. It would be helpful to include some comments on the potential role PTG plays in interferon or anti-pathogen pathways. For example: It is known that in some cell types, once infected with a virus they will switch from OXPHOS respiration to glycolysis. Maybe there is an anti-pathogen mechanism for PTG, and its expression is regulated for this. It would be interesting to discuss this, particularly as IRF4 and cGAMP play major roles in anti-bacterial immune pathways.

·         There are minor formatting issues throughout the review

Author Response

  1. In the introduction, you give a detailed overview of glycogen metabolism, however, I feel this information would be better supported with a figure describing the pathways.

Response: The enrichment of glucose induced glucokinase (GCK), which phosphorylated glucose to glucose-6-phosphate (G-6-P); It is used as a substrate for glycolysis or glycogen synthesis, depending on the nutritional status. The product pyruvate is further decarboxylated to acetyl coenzyme A and enters the tricarboxylic acid cycle (TCA) in mitochondria. Acetyl coenzyme A or malonyl coenzyme A synthesize fatty acids from scratch and further process them into TAG. In mammals, FA synthesis is catalyzed by acetyl CoA carboxylase (ACC) and fatty acid synthase (FAS).

  1. Line 76-78, this statement is a little confusing, and I feel following on from the previous sentence it would be better to state “As such the kinases and phosphatases are promising targets to modulate this response and such many inhibitors targeting these proteins have been developed”. I also think there is a lot of excellent points made about these inhibitors, and this information should also be summarised in a table to make it clear to the reader what they are targeting. This type of table would be very useful for the field.

Response: Thank you very much for your suggestion. A cohesive sentence has been added to the corresponding position of the article (line 75-77).

We summarized the drugs, targets and therapeutic fields of related protein phosphatase inhibitors. Here, we simply list the targets and functions of relevant phosphatase inhibitors. In the future, we may consult more literature to review the research progress of phosphatase inhibitors in detail. Our review mainly summarizes the research progress and application prospects of PTG, so we do not recommend this table to be included in the article.

Drug

Drug target

Field of treatment

MSI-1436[1]

PTP1B

Diabetes mellitus

IFB-088/ Sephin1[2, 3]

PPP1R15A

charcot-marie-tooth disease

Amyotrophic lateral sclerosis

AKB-9778[4]

VE-PTP

Diabetes retinopathy

Raphin1 [5]

PPP1R15B

Huntington disease

LB-100[6]

PP2A

Solid tumer

Recurrent glioblastoma

Stroke

Type 2 diabetes

PRL3-zumab[7]

PRL3

Advanced solid tumer

JAB-3068[8]

SHP2

Advanced solid tumer

TNO155[9]

SHP2

Advanced solid tumer

  1. Line 85 Protein phosphatase 1 should have the abbreviation after it (this happens in the second sentence.

Response: Thank you for your reminding. The abbreviations in the article have been modified (line 97).

  1. Line 208- You mention that PTG plays a role in glycogen synthesis in astrocyte cells. Is this specific to astrocyte cells in the CNS? Or are their other CNS cell types that PTG plays a role? If astrocytes are special in this sense, I think an explanation of what is currently know about astrocyte metabolism would be helpful- as they have alternate metabolism to neurons for example.

Response: Previous studies have shown that PTG is expressed in liver, muscle and adipocytes, and participates in glycogen metabolism. At present, the related studies believe that PTG is not limited to astrocytes, and other neurons also express PTG[10-12], but its role needs further study.

Astrocyte is the most widely distributed type of cells in mammalian brain and the largest type of glial cells. Its functions are complex and diverse, including that astrocytes have a spatial buffer effect on extracellular potassium ions, which can maintain the balance of ions around neurons. Metaglial cells are glutamate (Glu) and γ- Aminobutyric acid (GABA) metabolism site. In the role of neural development, astrocytes take nutrients from the blood to supply neurons and play a protective role. Astrocytes may participate in the process of trans synaptic signal transmission. There are some receptors on the cells and they are the main storage sites of glycogen in the brain. When neurons are highly active and the blood glucose provided through the blood brain barrier cannot meet the needs, the glycogen in the glial cells can be decomposed into glucose to provide energy for neurons under the effect of neurotransmitters. Astrocytes may also participate in a variety of neuropathological processes[13]. A description of astrocytes has been added to the article(line 254-261).

  1. Line 221- there is a reference missing

Response: Thank you for your suggestions. References have been added to the corresponding positions of the article(line 228-231).

  1. Line 223- I would make this a separate paragraph

Response: This part has been segmented (line 241-242).

  1. I think it’s interesting that there is a role for IRF4 and cGAMP in PTG expression. It would be helpful to include some comments on the potential role PTG plays in interferon or anti-pathogen pathways. For example: It is known that in some cell types, once infected with a virus they will switch from OXPHOS respiration to glycolysis. Maybe there is an anti-pathogen mechanism for PTG, and its expression is regulated for this. It would be interesting to discuss this, particularly as IRF4 and cGAMP play major roles in anti-bacterial immune pathways.

Response: Your proposal is very meaningful. Early relevant reports have pointed out that immune cells can improve their immune activity by processing glycogen, thereby helping to improve the effect of vaccines or reduce the severity of autoimmune diseases. Huang et al.[14] found that memory T cells regulate the formation and maintenance of memory through glycogen metabolism. In this metabolic model, glycogen mainly does not provide energy, but provides an effective way for cells to resist oxidation by producing NADPH and reduced glutathione. In their subsequent studies, the researcher also found that glycogen metabolism regulates the inflammatory phenotype of macrophages[15]. In addition, previous studies also found that glycogen synthase kinase 3 (GSK3) has become a key regulator of TLR signal, mediating the balance between proinflammatory and anti-inflammatory functions in the peripheral and central nervous systems. Through the use of primary microglial cell cultures and organ type hippocampal slices (OHSC), it was found that GSK3 inhibitors regulate inflammatory response by enhancing the down-regulation of pro-inflammatory genes and the up-regulation of anti-inflammatory function related genes, revealing that GSK3 is a key component of neuroinflammatory coordination and a target of neuroprotective strategies[16]. In view of this, PTG, an important glycogen metabolism phosphatase, may also participate in the immune inflammatory reaction. In the future, the role and potential value of PTG in immune inflammatory response need to be further explored.

Reviewer 3 Report

The review article written by Deng X, et al. summarizes the role of protein targeting to glycogen (PTG) in glucose and lipid metabolism. The manuscript first reviewed the function of protein phosphatase 1 (PP1) and then narrowed down to a member of PP1, PTG, focusing on the impacts of PTG on glucose and lipid homeostasis. While the topic is of great interest to the field, the discussion of PTG biological functions somewhat lacks in-depth narrative. There are also some points written in an ambiguous or subjective way. Overall, this manuscript is of potential interest to the readers. Some revisions are suggested:   

Major comments:

11. The overall layout of the manuscript is clear and easy to follow. Below are some structural suggestions.

1-1.  Introduction - last paragraph. It is suggested to expand it, since this is the most important paragraph in Introduction discussing the overall function of PP1 and/or PTG.

1-2.  Lines 105 – 108 vs Lines 114-118. The 2 are somehow repeated contents. Please just keep one.

1-3.  “5. PTG and lipid metabolism”. There is just one long paragraph (Line 313 to 377). It is suggested to separate it by subtopic as did in “4. PTG and glucose metabolism”.

22. Ambiguous and subjective expression are something the authors may want to avoid. Below are some examples. Please note that it is not limited to the listed ones.

2-1. Line 32. “factors such as genes and excess nutrition”. What “genes”?

2-2. Line 42. “Human cells mainly reply on the liver…” I assume that the phenomena are not limited to the humans.

2-3. Line 315. “Hepatic de novo lipogenesis …helps hepatocytes store and secrete lipids”. De novo lipogenesis (DNL) defines as an endogenous process that excess dietary nutrients are converted to fatty acids. Secretion of lipids are referred to VLDL secretion and is not involved in DNL.

2-4. Line 335. “The current data show that…”. Which data? It is confusing.

2-5. Conclusions. “The review increases people’s understanding…; expands people’s understanding…, and provides theoretical references…”. I would suggest to focus on the contents that have been discussed in the manuscript and summarize the major points.   

33. Although the manuscript lists a subtitle as “PTG on lipid metabolism”. The authors did not really explain the exact role of PTG in lipid metabolism. The long paragraph is mixed with background introduction of basic concepts about lipid metabolism, findings pointing some links between glucose and lipid homeostasis, and a few studies investigating PTG in high fat models. Discuss these points one by one and clearly stating the boundaries would benefit reading.

44. The manuscript would benefit from English editing.

Minor comments:

11. Title. It is suggested to include the full name of PTG (protein targeting to glycogen). If there is a link between two things, there is a relationship between them. However, to my understanding, glucose metabolism and lipid metabolism are the 2 aspects that PTG plays a key role in, not like PTG-medicated glucose metabolism causes or affects lipid homeostasis or vice versa. An alternative name, such as “Protein targeting to glycogen (PTG): an promising player in glucose and lipid metabolism”, is suggested.

d2. List full name of an abbreviation the first time it is listed. Line 114 shows both full name and abbreviation of "PTG". However, it has already been mentioned multiple time among Lines 95-110. Another example is "GL" in Line 171. What does it stand for?

Author Response

1-1 Introduction - last paragraph. It is suggested to expand it, since this is the most important paragraph in Introduction discussing the overall function of PP1 and/or PTG.

Response: Thank you for your advice. We have expanded the introduction as suggested (line 97-105).

1-2.  Lines 105 – 108 vs Lines 114-118. The 2 are somehow repeated contents. Please just keep one.

Response: Thanks for your reminder, one of them has been deleted.

1-3.  “5. PTG and lipid metabolism”. There is just one long paragraph (Line 313 to 377). It is suggested to separate it by subtopic as did in “4. PTG and glucose metabolism”.

Response: This part has been described in sections according to your suggestions (line347,359).

  1. Ambiguous and subjective expression are something the authors may want to avoid. Below are some examples. Please note that it is not limited to the listed ones.

2-1. Line 32. “factors such as genes and excess nutrition”. What “genes”?

Response: The "gene" here should refer to hormones, which may be caused by confusion due to problems in expression, and has been modified accordingly in the article (line 33).

2-2. Line 42. “Human cells mainly reply on the liver…” I assume that the phenomena are not limited to the humans.

Response: The sentence has been revised accordingly (line 43).

2-3. Line 315. “Hepatic de novo lipogenesis …helps hepatocytes store and secrete lipids”. De novo lipogenesis (DNL) defines as an endogenous process that excess dietary nutrients are converted to fatty acids. Secretion of lipids are referred to VLDL secretion and is not involved in DNL.

Response: This sentence has been modified (line 333-334).

2-4. Line 335. “The current data show that…”. Which data? It is confusing.

Response: Some contents of “4. PTG and glucose metabolism” have been modified accordingly, and relevant contents have been revised.

2-5. Conclusions. “The review increases people’s understanding…; expands people’s understanding…, and provides theoretical references…”. I would suggest to focus on the contents that have been discussed in the manuscript and summarize the major points.   

Response: The summary has been revised accordingly (line 420-426).

  1. Although the manuscript lists a subtitle as “PTG on lipid metabolism”. The authors did not really explain the exact role of PTG in lipid metabolism. The long paragraph is mixed with background introduction of basic concepts about lipid metabolism, findings pointing some links between glucose and lipid homeostasis, and a few studies investigating PTG in high fat models. Discuss these points one by one and clearly stating the boundaries would benefit reading.

Response: This part has been revised, and the relationship between PTG and lipid metabolism has been discussed one by one. I hope this change is conducive to better understanding the progress of PTG in lipid metabolism (line347,359).

  1. The manuscript would benefit from English editing.

Response: With regards to English language copy editing, an English language native speaker assisted us to proofread the manuscript. Of course, we are willing to cooperate with the editorial department in the follow-up process.

Minor comments:

  1. Title. It is suggested to include the full name of PTG (protein targeting to glycogen). If there is a link between two things, there is a relationship between them. However, to my understanding, glucose metabolism and lipid metabolism are the 2 aspects that PTG plays a key role in, not like PTG-medicated glucose metabolism causes or affects lipid homeostasis or vice versa. An alternative name, such as “Protein targeting to glycogen (PTG): an promising player in glucose and lipid metabolism”, is suggested.

Response: Your suggestion is very good, and we were once troubled by the title. Changes to abbreviations have been noted in the text (line 91).

d2. List full name of an abbreviation the first time it is listed. Line 114 shows both full name and abbreviation of "PTG". However, it has already been mentioned multiple time among Lines 95-110. Another example is "GL" in Line 171. What does it stand for?

Response: We are very sorry for the misunderstanding caused by the format error. PP1 has four glycogen targeting subunits: GM, GL, PPP1R6 and glycogen target protein (PTG). GL is also known as PPP1R3.

Round 2

Reviewer 3 Report

The authors have addressed the concerns. No further comments.